# WEAKLY-SUPERVISED HOI DETECTION VIA PRIOR-GUIDED BI-LEVEL REPRESENTATION LEARNING

**Bo Wan** [1]* **Yongfei Liu** [2]*, **Desen Zhou** [2], **Tinne Tuytelaars** [1], **Xuming He** [2,3]

[1] KU Leuven, Leuven, Belgium;     [2] ShanghaiTech University, Shanghai, China
[3] Shanghai Engineering Research Center of Intelligent Vision and Imaging

{bwan,tinne.tuytelaars}@esat.kuleuven.be
{liuyf3,zhouds,hexm}@shanghaitech.edu.cn

## ABSTRACT

Human object interaction (HOI) detection plays a crucial role in human-centric scene understanding and serves as a fundamental building-block for many vision tasks. One generalizable and scalable strategy for HOI detection is to use weak supervision, learning from image-level annotations only. This is inherently challenging due to ambiguous human-object associations, large search space of detecting HOIs and highly noisy training signal. A promising strategy to address those challenges is to exploit knowledge from large-scale pretrained models (e.g., CLIP), but a direct knowledge distillation strategy (Liao et al., 2022) does not perform well on the weakly-supervised setting. In contrast, we develop a CLIP-guided HOI representation capable of incorporating the prior knowledge at both image level and HOI instance level, and adopt a self-taught mechanism to prune incorrect human-object associations. Experimental results on HICO-DET and V-COCO show that our method outperforms the previous works by a sizable margin, showing the efficacy of our HOI representation.

## 1 INTRODUCTION

Human object interaction detection aims to simultaneously localize the human-object regions in an image and to classify their interactions, which serves as a fundamental building-block in a wide range of tasks in human-centric artificial intelligence, such as human activity recognition (Heilbron et al., 2015; Tina et al., 2021), human motion tracking (Wafae et al., 2019; Nishimura et al., 2021) and anomalous behavior detection (Liu et al., 2018; Pang et al., 2020).

Usually, HOI detection adopts a supervised learning paradigm (Gupta & Malik, 2015; Chao et al., 2018; Wan et al., 2019; Gao et al., 2020; Zhang et al., 2021c). This requires detailed annotations (i.e. human and object bounding boxes and their interaction types) in the training stage. However, such HOI annotations are expensive to collect and prone to labeling errors. In contrast, it is much easier to acquire image-level descriptions of target scenes. Consequently, a more scalable strategy for HOI detection is to learn from weak annotations at the image level, known as weakly-supervised HOI detection (Zhang et al., 2017). Learning under such weak supervision is particularly challenging mainly due to the lack of accurate visual-semantic associations, large search space of detecting HOIs and highly noisy training signal from only image level supervision.

Most existing works (Zhang et al., 2017; Baldassarre et al., 2020; Kumaraswamy et al., 2021) attempt to tackle the weakly-supervised HOI detection in a Multiple Instance Learning (MIL) framework (Ilse et al., 2018). They first utilize an object detector to generate human-object proposals and then train an interaction classifier with image-level labels as supervision. Despite promising results, these methods suffer from several weaknesses when coping with diverse and fine-grained HOIs. Firstly, they usually rely on visual representations derived from the external object detector, which mainly focus on the semantic concepts of the objects in the scene and hence are insufficient for capturing the concept of fine-grained interactions. Secondly, as the image-level supervision tends to ignore the imbalance in HOI classes, their representation learning is more susceptible to the dataset bias and dominated by frequent interaction classes. Finally, these methods learn the HOI concepts from a candidate set generated by pairing up *all* the human and object proposals, which is highly noisy and often leads to erroneous human-object associations for many interaction classes.

---

*Equal Contribution. Code is available at https://github.com/bobwan1995/Weakly-HOI.

To address the aforementioned limitations, we introduce a new weakly-supervised HOI detection strategy. It aims to incorporate the prior knowledge from pretrained foundation models to facilitate the HOI learning. In particular, we propose to integrate CLIP (Radford et al., 2021b), a large-scale vision-language pretrained model. This allows us to exploit the strong generalization capability of the CLIP representation for learning a better HOI representation under weak supervision. Compared to the representations learned by the object detector, the CLIP representations are inherently less object-centric, hence more likely to incorporate also aspects about the human-object interaction, as evidenced by Appendix A. Although a few works have successfully exploited CLIP for supervised HOI detection in the past, experimentally we find they do not perform well in the more challenging weakly-supervised setting (c.f. Appendix.B). We hypothesize this is because they only transfer knowledge at image level, and fail without supervision at the level of human-object pairs.

To this end, we develop a CLIP-guided HOI representation capable of incorporating the prior knowledge of HOIs at two different levels. First, at the image level, we utilize the visual and linguistic embeddings of the CLIP model to build a global HOI knowledge bank and generate image-level HOI predictions. In addition, for each human-object pair, we enrich the region-based HOI features by the HOI representations in the knowledge bank via a novel attention mechanism. Such a bi-level framework enables us to exploit the image-level supervision more effectively through the shared HOI knowledge bank, and to enhance the interaction feature learning by introducing the visual and text representations of the CLIP model.

We instantiate our bi-level knowledge integration strategy as a modular deep neural network with a global and local branch. Given the human-object proposals generated by an off-the-shelf object detector, the global branch starts with a *backbone network* to compute image feature maps, which are used by a subsequent *HOI recognition network* to predict the image-wise HOI scores. The local branch builds a *knowledge transfer network* to extract the human-object features and augment them with the CLIP-guided knowledge bank, followed by a *pairwise classification network* to compute their relatedness and interaction scores [1]. The relatedness scores are used to prune incorrect human-object associations, which mitigates the issue of noisy proposals. Finally, the outputs of the two branches are fused to generate the final HOI scores.

To train our HOI detection network with image-level annotations, we first initialize the backbone network and the HOI knowledge bank from the CLIP encoders, and then train the entire model in an end-to-end manner. In particular, we devise a novel multi-task weak supervision loss consisting of three terms: 1) an image-level HOI classification loss for the global branch; 2) an MIL-like loss for the interaction scores predicted by the local branch, which is defined on the aggregate of all the human-object pair predictions; 3) a self-taught classification loss for the relatedness of each human-object pair, which uses the interaction scores from the model itself as supervision.

We validate our methods on two public benchmarks: HICO-DET (Chao et al., 2018) and V-COCO (Gupta & Malik, 2015). The empirical results and ablative studies show our method consistently achieves state-of-the-art performance on all benchmarks. In summary, our contributions are three-fold: (i) We exploit the CLIP knowledge to build a prior-enriched HOI representation, which is more robust for detecting fine-grained interaction types and under imbalanced data distributions. (ii) We develop a self-taught relatedness classification loss to alleviate the problem of mis-association between human-object pairs. (iii) Our approach achieves state-of-the-art performance on the weakly-supervised HOI detection task on both benchmarks.

## 2 RELATED WORKS

**HOI detection:** Most works on supervised HOI detection can be categorized in two groups: two-stage and one-stage HOI detection. Two-stage methods first generate a set of human-object proposals with an external object detector, then classify their interactions. They mainly focus on exploring additional human pose information (Wan et al., 2019; Li et al., 2020a; Gupta et al., 2019), pairwise relatedness (Li et al., 2019a; Zhou et al., 2020) or modeling relations between object and human (Gao et al., 2020; Zhang et al., 2021c; Ulutan et al., 2020; Zhou & Chi, 2019), to enhance the HOI representations. One-stage methods predict human & object locations and their interaction types simultaneously in an end-to-end manner, which are currently dominated by transformer-based architectures (Carion et al., 2020; Kim et al., 2022; Dong et al., 2022; Zhang et al., 2021a;b).

---

[1] Relatedness indicates whether a human-object pair has a relation, and interaction scores are multi-label scores on the interaction space.

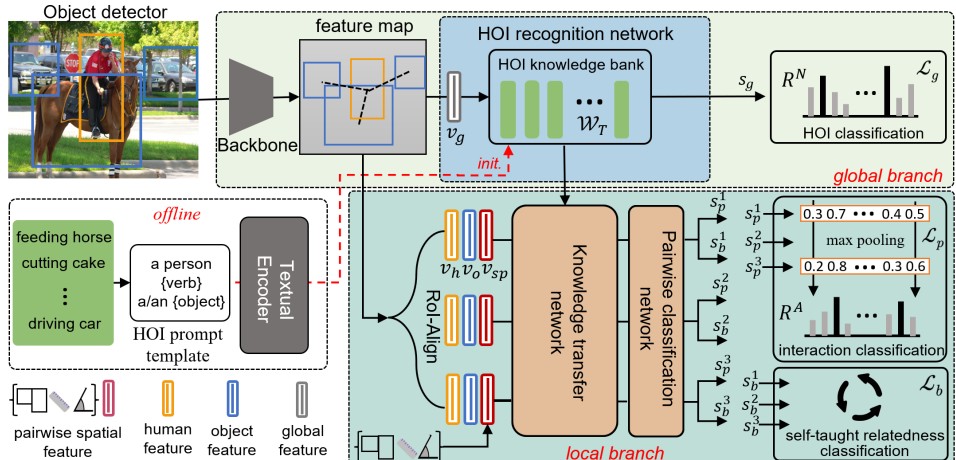

Figure 1: **Model Overview:** There are four modules in our network: a backbone Network, an HOI recognition network, a knowledge transfer network and a pairwise classification network.

Supervised methods show superior performance, but require labor-intensive HOI annotations that are infeasible to obtain in many scenarios. Thus, in this work we focus on HOI detection under weak supervision.

**Weakly-supervised HOI detection:** Weakly-supervised HOI detection aims to learn instance-level HOIs with *only image-level annotations*. (Prest et al., 2011) learns a set of binary action classifiers based on detected human-object pairs, where human proposal is obtained from a part-based human detector and object is derived from the relative position with respect to the human. PPR-FCN (Zhang et al., 2017) employs a parallel FCN to perform pair selection and classification. Explainable-HOI (Baldassarre et al., 2020) adopts graph nets to capture relations for better image-level HOI recognition, and uses backward explanation for instance-level HOI detection. MX-HOI (Kumaraswamy et al., 2021) proposes a momentum-independent learning strategy to utilize strong & weak labels simultaneously. AlignFormer (Kilickaya & Smeulders, 2021) proposes an align layer in transformer framework, which utilizes geometric & visual priors to generate pseudo alignments for training. Those methods focus on learning HOIs with advanced network structures or better pseudo alignments. However, they still suffer from noisy human-object associations and ambiguous interaction types. To address those challenges, we exploit prior knowledge from CLIP to build a discriminative HOI representations.

**Knowledge exploitation of pretrained V&L models:** Recently, CLIP (Radford et al., 2021a) model has demonstrated strong generalization to various downstream tasks (Ghiasi et al., 2021; Du et al., 2022; Gu et al., 2021). Some works also explore CLIP knowledge in supervised HOI detection, e.g., CATN (Dong et al., 2022) initializes the object query with category-aware semantic information from CLIP text encoder, and GEN-VLTK (Liao et al., 2022) employs image feature distillation and classifier initialization with HOI prompts. However, they only exploit CLIP knowledge at a coarse level and require detailed annotations of human-object pairs. It is non-trivial to extend such strategies to the weak supervision paradigm due to highly noisy training signals. In our work, we build a deep connection between CLIP and HOI representation by incorporating the prior knowledge of HOIs at both image and HOI instance levels.

## 3 METHOD

### 3.1 PROBLEM SETUP AND METHOD OVERVIEW

**Problem setup**  Given an input image $I$, the task of weakly-supervised HOI detection aims to localize and recognize the human-object interactions, while only the corresponding image-level HOI categories are available for training. Formally, we aim to learn a HOI detector $\mathcal{M}$, which takes an image $I$ as input and generates a set of tuples $\mathcal{O} = \{(\mathbf{x}_h, \mathbf{x}_o, c_o, a_{h,o}, R^a_{h,o})\}$, i.e., $\mathcal{O} = \mathcal{M}(I)$. Here each tuple indicates a HOI instance, in which $\mathbf{x}_h, \mathbf{x}_o \in \mathbb{R}^4$ represent human and object bounding boxes, $c_o \in \{1, ..., C\}$ is the object category, $a_{h,o} \in \{1, ..., A\}$ denotes the interaction class associated with $\mathbf{x}_h$ and $\mathbf{x}_o$, and $R^a_{h,o} \in \mathbb{R}$ is the HOI score. For the weakly-supervised setting,

each training image is annotated with a set of HOI categories $\mathcal{R} = \{r^*\}$ at the image level only, where $r^* \in \{1, ..., N\}$ is an index to a combination of ground-truth object category $c^*$ and interaction category $a^*$, and $N$ denotes the number of all possible HOI combinations defined on the dataset.

**Method Overview** As we lack supervision for the HOI locations, we adopt a typical *hypothesize-and-recognize* strategy (Zhang et al., 2017; Baldassarre et al., 2020; Kumaraswamy et al., 2021) for HOI detection: first we generate a set of human and object proposals with an off-the-shelf object detector (Ren et al., 2015) and then predict the interaction class for all human-object combinations.

Unlike other methods, we do not re-use the feature maps of the object or human detector - we only keep the bounding boxes. Instead, we learn a new representation optimized for the HOI task. This is challenging under the weak setting as the model learning is noisy, but feasible by leveraging the rich semantic knowledge from a pretrained large-scale multimodal model, like CLIP. However, the naive knowledge integration strategies for supervised setting fail when directly applied in the weak setting, as evidenced by our experiments in Appendix.B

Our framework adopts two philosophies to address the challenges in the weakly-supervised HOI task: the first is to integrate the prior knowledge into discriminative representation learning, and the second is to suppress noise in learning. For the first philosophy, we utilize the prior knowledge from CLIP to guide the representation learning in both global image-level and fine-grained human-object pairs, which is instantiated by a bi-level knowledge integration strategy. For the second philosophy, we adopt an effective self-taught learning mechanism to suppress the irrelevant pairs.

We instantiate the bi-level knowledge integration strategy with a two-branch deep network. Our detection pipeline starts with a set of human proposals with detection scores $\{(\mathbf{x}_h, s_h)\}$, and object proposals with their categories and detection scores $\{(\mathbf{x}_o, c_o, s_o)\}$. Then, the global branch performs image-level HOI recognition by utilizing a CLIP-initialized HOI knowledge bank as a classifier. This allows us to exploit both visual and text encoders from CLIP to generate better HOI representations. In parallel, for each human-object pair $(\mathbf{x}_h, \mathbf{x}_o)$, the local branch explicitly augments the pairwise HOI features with the HOI knowledge bank to then identify their relatedness and interaction classes.

To train our model, we use a multi-task loss, which incorporates a HOI recognition loss defined on image-wise HOIs for the visual encoder and knowledge bank finetuning, and a self-taught relatedness classification for suppressing the background human-object associations, on top of the standard MIL-based loss. We first present model details in Sec.3.2, followed by the training strategy in Sec.3.3.

## 3.2 MODEL DESIGN

Now we introduce our bi-level knowledge integration strategy, where the aim is to exploit CLIP textual embeddings of HOI labels as a HOI knowledge bank for the HOI representation learning, and to transfer such knowledge both at image level as well as at the level of human-object pairs for interaction predictions. Specifically, as shown in Fig. 1, our network consists of a global branch and a local branch. The global branch includes a backbone network (Sec.3.2.1) that extracts image features, and a HOI recognition network (Sec.3.2.2) that uses a HOI knowledge bank based on CLIP to predict image-level HOI scores. For each human-object proposal generated by an off-the-shelf object detector, the local branch employs a knowledge transfer network (Sec.3.2.3) to compute its feature representation with enhancement from the HOI knowledge bank, and a pairwise classification network (Sec.3.2.4) to compute their relatedness and interaction scores. Finally, we generate the final HOI detection scores by combining global HOI scores with local predictions (Sec. 3.2.5).

**HOI Knowledge Bank Generation** CLIP builds a powerful vision-language model by pretraining on large-scale image-text pairs. It consists of a visual encoder $\mathcal{F}_V$ and textual encoder $\mathcal{F}_T$, mapping both visual and textual inputs to a shared latent space. Here, we exploit CLIP to generate a HOI knowledge bank. We take a similar prompt strategy as in CLIP, adopting a common template 'a person {*verb*} a/an {*object*}' to convert HOI labels into text prompts (e.g., converting 'drive car' to 'a person driving a car'). Then we input the sentences into the CLIP textual encoder $\mathcal{F}_T$ to initialize the HOI knowledge bank $\mathcal{W}_T \in \mathbb{R}^{N \cdot D}$, with $D$ denoting the feature dimension. One can think of $\mathcal{W}_T$ as a set of 'prototypes' in feature space, one for each HOI in the dataset.

### 3.2.1 GLOBAL BRANCH: BACKBONE NETWORK

To incorporate CLIP for feature extraction, we initialize the backbone network (e.g., a ResNet-101 (He et al., 2016)) with CLIP's visual encoder $\mathcal{F}_V$ to generate a feature map $\boldsymbol{\Gamma}$ for the input image $I$. We further compute a global feature vector $v_g \in \mathbb{R}^D$ with self-attention operation (Radford et al., 2021b).

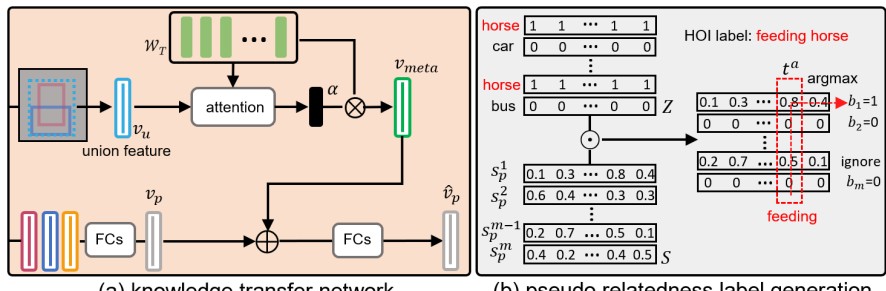

Figure 2: The **knowledge transfer network** explicitly transfers the discriminative relation-level semantic knowledge derived from CLIP to the pairwise HOI representations. **Pseudo relatedness label generation** uses the pairwise interaction scores to generate the pseudo association labels for self-taught relatedness classification

### 3.2.2 GLOBAL BRANCH: HOI RECOGNITION NETWORK

We perform an image-wise HOI recognition task with the HOI knowledge bank $\mathcal{W}_T$. We obtain global HOI scores $s_g \in \mathbb{R}^N$ by computing the inner product between the image feature $v_g$ and the knowledge bank $\mathcal{W}_T$: $s_g = \mathcal{W}_T \times v_g$, where $\times$ is matrix multiplication. This has the effect of adapting the visual encoder and knowledge bank parameters to the HOI recognition task, fully taking advantage of the knowledge from CLIP.

### 3.2.3 LOCAL BRANCH: KNOWLEDGE TRANSFER NETWORK

Given the CLIP-initialized visual encoder, a standard HOI representation can be formed by concatenating the human and object appearance features along with their spatial encoding. However, even after the finetuning as described above, such a representation still mainly focuses on object-level semantic cues rather than relation-level concepts. In this module, we explicitly exploit the HOI knowledge bank $\mathcal{W}_T$ to learn a local relation-specific HOI representation. To achieve this, we propose an attention-based architecture as shown in Fig.2(a).

Specifically, for each human proposal $\mathbf{x}_h$ and object proposal $\mathbf{x}_o$, we use RoI-Align (He et al., 2017) to crop the feature maps from $\Gamma$ followed by a self-attention operation to compute their appearance features $v_h, v_o \in \mathbb{R}^D$. Then we compute a spatial feature $v_{sp}$ by encoding the relative positions of their bounding boxes $(\mathbf{x}_h, \mathbf{x}_o)$ [2]. The holistic HOI representation $v_p \in \mathbb{R}^D$ is an embedding of the human and object appearance features and their spatial feature, i.e., $v_p = \mathcal{F}_E([v_h; v_o; v_{sp}])$, where $[;]$ is the concatenation operation and $\mathcal{F}_E$ is a multi-layer perceptron (MLP).

To enhance relation-level concepts, we further compute its union region $\mathbf{x}_u \in \mathbb{R}^4$ (see Fig. 2a) and extract the corresponding appearance feature $v_u \in \mathbb{R}^D$ via RoI-align over the feature map $\Gamma$. The union region is important as it encodes relational context cues, but it potentially also contains a large amount of background that is noisy for model learning. We thus devise an attention module that is similar in design to the HOI recognition network, but uses the union feature $v_u$ as query to extract a meta-embedding $v_{meta} \in \mathbb{R}^D$ from the HOI knowledge bank $\mathcal{W}_T$. The final HOI representation $\hat{v}_p \in \mathbb{R}^D$ is built by fusing the holistic representation $v_p$ and $v_{meta}$ with a MLP $\mathcal{F}_K$.

$$\alpha = Softmax(\mathcal{W}_T \times v_u); \quad v_{meta} = \alpha^{\intercal} \times \mathcal{W}_T; \quad \hat{v}_p = \mathcal{F}_K(v_p + v_{meta}). \quad (1)$$

Here $\alpha \in \mathbb{R}^N$ is the normalized attention weight and $\intercal$ is the transpose operation. $v_{meta}$ encodes a discriminative representation from CLIP and facilitates feature sharing between HOI classes.

### 3.2.4 LOCAL BRANCH: PAIRWISE CLASSIFICATION NETWORK

Given the relation-aware HOI representation $\hat{v}_p$, our final module performs a coarse-level classification on human-object association and a fine-level classification for interaction recognition. Specifically, we use two MLPs $\mathcal{F}_P$ and $\mathcal{F}_B$ to predict the interaction scores $s_p \in \mathbb{R}^A$ and the relatedness score $s_b \in \mathbb{R}$ for each human-object pair:

$$s_p = \mathcal{F}_P(\hat{v}_p); \quad s_b = \mathcal{F}_B(\hat{v}_p) \quad (2)$$

---

[2]For details c.f. the appendix C

To train the model under weak supervision (see Sec. 3.3), we further aggregate the pairwise interaction scores into image-level interaction scores . Assume we have $M$ pairs of human-object proposals for a given image, and denote the interaction scores for the $m$-th pair as $s_p^m$. We first concatenate all the interaction scores to compose a bag $S = [s_p^1; ...; s_p^M] \in \mathbb{R}^{M \cdot A}$, then we maximize over all pairs to obtain the image-wise interaction scores: $\tilde{s}_p = \max_m S$.

### 3.2.5 MODEL INFERENCE

During model inference, we do not use the local interaction scores $s_p$ directly. Instead, we normalize $S$ with a $Softmax$ operation defined on all pairs: $\bar{S} = Softmax_m(S)$, and then compute the normalized pairwise interaction scores $e_p = \sigma(\tilde{s}_p) \cdot \bar{s}_p$, where $\bar{s}_p$ is a row from $\bar{S}$ and $\sigma$ is $Sigmoid$ function. This has the effect of measuring the contribution of a given pair, in case of multiple pairs in an image share the same interaction.

The final interaction score $s_{h,o}^a$ for human-object pair $(\mathbf{x}_h, \mathbf{x}_o)$ combines multiple scores, including the global HOI scores $s_g$, the normalized pairwise interaction scores $e_p$, and the relatedness score $s_b$. The overall HOI score $R_{h,o}^a$ is a combination of the interaction score and the object detection scores.

$$s_{h,o}^a = \sigma(s_g^{a,c_o}) \cdot e_p^a \cdot \sigma(s_b); \quad R_{h,o}^a = (s_h \cdot s_o)^\gamma \cdot s_{h,o}^a \tag{3}$$

where $s_g^{a,c_o}$ is the HOI score corresponding to $a$-th interaction and $c_o$-th object category in $s_g$, $e_p^a$ is the score of $a$-th interaction in $e_p$, and $\gamma$ is a hyper-parameter to balance the scores (Zhang et al., 2021c; Li et al., 2019b).

### 3.3 LEARNING WITH WEAK SUPERVISION

To train our deep network in a weakly supervised setting, we use a multi-task loss defined on three different levels. Specifically, our overall loss function $\mathcal{L}$ consists of three terms: i) an image-wise HOI recognition loss $\mathcal{L}_g$ to adapt CLIP features to the task of human-object interaction detection; ii) a pairwise interaction classification loss $\mathcal{L}_p$ to guide the knowledge transfer towards fine-grained relation-aware representations; and iii) a self-taught relatedness classification loss $\mathcal{L}_b$ to prune non-interacting human-object combinations. Formally, the overall loss is written as:

$$\mathcal{L} = \mathcal{L}_g + \mathcal{L}_p + \mathcal{L}_b \tag{4}$$

**Image-wise HOI recognition loss $\mathcal{L}_g$:** Given the HOI scores $s_g$ and ground-truth HOI categories $\mathcal{R}$, $\mathcal{L}_g$ is a standard binary cross-entropy loss for multi-label classification: $\mathcal{L}_g = L_{BCE}(s_g, \mathcal{R})$.

**Pairwise interaction classification loss $\mathcal{L}_p$:** We adopt a MIL strategy that first aggregates the pairwise interaction scores and supervises this with image-level interaction labels as $\mathcal{A} = \{a^*\}$. Given the image-wise interaction scores $\tilde{s}_p$, $\mathcal{L}_p$ is a standard binary cross-entropy loss for multi-label classification as: $\mathcal{L}_p = L_{BCE}(\tilde{s}_p, \mathcal{A})$.

**Self-taught relatedness classification loss $\mathcal{L}_b$:** As human-object associations are not annotated, we devise a novel *pseudo relatedness label generation mechanism* for training a self-taught binary classifier to identify valid human-object associations. Specifically, we observe that the human-object pairs with confident interaction scores are often associated after a short period of initial training without self-taught classification loss. Motivated by this, we use the interaction scores $s_p$ from the model under training to supervise the relatedness classification.

Concretely, we generate pseudo labels $\mathcal{B} = \{b_1, ..., b_M\}$ for all human-object pairs in an image, where $b_m \in \{0, 1\}$ indicates the relatedness for the $m$-th combination. To this end, as illustrated in Fig.2(b), we first propose a binary mask $Z \in \{0, 1\}^{M \cdot A}$ for all interaction scores $S$ with respect to the ground-truth object categories $\mathcal{C} = \{c^*\}$. For each human-object pair where the object label $c_o$ is included in $\mathcal{C}$, we consider it as a potential interactive combination and thus assign the corresponding row in $Z$ as 1, and other rows as 0. For the latter, we also immediately set $b_m = 0$. Then we generate pairwise scores $t^a \in \mathbb{R}^M$ for each ground-truth interaction $a^*$ by selecting the corresponding row from $S \odot Z$. The pseudo label for the pair with the highest score is assigned as 1, i.e., $m_a = \arg\max_m t^a$ and $b_{m_a} = 1$. We only select one positive pair[3] for each $a^*$. Finally, $\mathcal{L}_b$ is defined as a binary cross-entropy loss: $\mathcal{L}_b = \sum_m L_{BCE}(s_b^m, b_m)$, where $s_b^m$ is the relatedness score for the $m$-th pair.

---

[3]We also explore top-K selection in Appendix F

Table 1: mAP comparison on HICO-DET and V-COCO test set. - denotes the results are not available. * stands for the method we re-evaluate with the correct evaluation protocol (see Appendix.I for details) and †means our re-implementation. For V-COCO, all object detectors are pretrained on MSCOCO dataset by default, and details about the evaluation metrics APS1&2 c.f. Appendix H. IN-1K denotes ImageNet with 1000 classes.

| Methods | Backbone | Detector | HICO-DET (%) | | | V-COCO (%) | |
|---|---|---|---|---|---|---|---|
| | | | Full | Rare | Non-Rare | $AP_{role}^{S1}$ | $AP_{role}^{S2}$ |
| *supervised* | | | | | | | |
| iCAN (Gao et al., 2018) | RN50 (IN-1K&COCO) | FRCNN (COCO) | 14.84 | 10.45 | 16.15 | 45.30 | 52.40 |
| PMFNet (Wan et al., 2019) | RN50-FPN (IN-1K&COCO) | FRCNN (COCO) | 17.46 | 15.56 | 18.00 | 52.00 | - |
| TIN (Li et al., 2019b) | RN50-FPN (IN-1K&COCO) | FRCNN (COCO) | 17.22 | 13.51 | 18.32 | 47.80 | 54.20 |
| DJ-RN (Li et al., 2020a) | RN50 (IN-1K&COCO) | FRCNN (COCO) | 21.34 | 18.53 | 21.18 | 53.30 | 60.30 |
| IDN (Li et al., 2020b) | RN50 (IN-1K&COCO) | FRCNN (HICO-DET) | 26.29 | 22.61 | 27.39 | 53.30 | 60.30 |
| SCG (Zhang et al., 2021c) | RN50-FPN (IN-1K&HICO-DET) | FRCNN (HICO-DET) | 31.33 | 24.72 | 33.31 | 54.20 | 60.90 |
| HOTR (Kim et al., 2021) | RN50+Transformer (IN-1K&COCO) | DETR (HICO-DET) | 25.10 | 17.34 | 27.42 | 55.20 | 64.40 |
| QPIC (Tamura et al., 2021) | RN101+Transformer (IN-1K&COCO) | DETR (COCO) | 29.90 | 23.92 | 31.69 | 58.30 | 60.70 |
| CATN (Dong et al., 2022) | RN50+Transformer (IN-1K&HICO-DET&COCO) | DETR (HICO-DET) | 31.86 | 25.15 | 33.84 | 60.10 | - |
| MSTR (Kim et al., 2022) | RN50 + Transformer (IN-1K&COCO) | DETR(HICO-DET) | 31.17 | 25.31 | 33.92 | 62.00 | 65.20 |
| DisTr (Zhou et al., 2022) | RN50+Transformer (IN-1K&COCO) | DETR (HICO-DET) | 31.75 | 27.45 | 33.03 | 66.20 | 68.50 |
| SSRT (Iftekhar et al., 2022) | R101+Transformer (IN-1K&COCO) | DETR (COCO) | 31.34 | 24.31 | 33.32 | 65.00 | 67.10 |
| GEN-VLKT (Liao et al., 2022) | RN101+Transformer (IN-1K&HICO-DET) | DETR (HICO-DET) | 34.95 | 31.18 | 36.08 | 63.58 | 65.93 |
| *between supervised & weakly-supervised setting, learning with image-level HOIs and box annotations* | | | | | | | |
| AlignFormer (Kilickaya & Smeulders, 2021) | RN101+Transformer (IN-1K&HICO-DET) | DETR (HICO-DET) | 20.85 | 18.23 | 21.64 | 15.82 | 16.34 |
| *weakly-supervised* | | | | | | | |
| Explanation-HOI* (Baldassarre et al., 2020) | ResNeXt101 (IN-1K&COCO) | FRCNN (COCO) | 10.63 | 8.71 | 11.20 | - | - |
| MX-HOI (Kumaraswamy et al., 2021) | RN101 (IN-1K&COCO) | FRCNN (COCO) | 16.14 | 12.06 | 17.50 | - | - |
| PPR-FCN† (Zhang et al., 2017) | RN50 (CLIP dataset) | FRCNN (COCO) | 17.55 | 15.69 | 18.41 | - | - |
| *ours* | RN50 (CLIP dataset) | FRCNN (COCO) | 22.89 | 22.41 | 23.03 | 42.97 | 48.06 |
| *ours* | RN101 (CLIP dataset) | FRCNN (COCO) | 25.70 | 24.52 | 26.05 | 44.74 | 49.97 |

# 4 EXPERIMENTS

## 4.1 EXPERIMENTAL SETUP

**Datasets:** We benchmark our model on two public datasets: HICO-DET and V-COCO. HICO-DET consists of 47776 images (38118 for training and 9658 for test). It has $N = 600$ HOI categories, which are composed of $C = 80$ common objects (the same as MSCOCO (Lin et al., 2014)) and $A = 117$ unique interaction categories. V-COCO is a subset of MSCOCO, consisting of 2533 images for training, 2867 for validation and 4946 for test. It has 16199 human instances, each annotated with binary labels for $A = 26$ interaction categories.

**Evaluation Metric:** Following (Chao et al., 2015), we use mean average precision (mAP) to evaluate HOI detection performance. A human-object pair is considered as positive when both predicted human and object boxes have at least 0.5 IoU with their ground-truth boxes, and the HOI class is classified correctly.

## 4.2 IMPLEMENTATION DETAILS

We use an off-the-shelf Faster R-CNN (Ren et al., 2015) pretrained on MSCOCO to generate at most 100 object candidates for each image. For V-COCO, it is worth noting that we train the object detector by removing the images in MSCOCO that overlap with V-COCO to prevent information leakage. The backbone network is initialized with the visual encoder from CLIP-RN101 model and the feature dimension $D = 1024$.

For model learning, we set the detection score weight $\gamma = 2.8$ as default by following previous works (Zhang et al., 2021c; Li et al., 2019b), then optimize the entire network with AdamW and an initial learning rate of 1e-5 for backbone parameters and 1e-4 for others. We detach the parameters of the knowledge bank on the local branch for better model learning. We train up to 60K iterations with batch-size 24 in each on 4 NVIDIA 2080TI GPUs, and decay the learning rate by 10 times in 12K and 24K iteration.

## 4.3 QUANTITATIVE RESULTS

For **HICO-DET** (Tab.1), our approach outperforms the previous state of the arts on the weakly supervised setting by a clear margin, achieving 22.89 mAP with ResNet-50 and 25.70 mAP with ResNet-101 as backbone. For a fair comparison, we also re-implement PPR-FCN with CLIP visual encoder. The results show that we still outperform PPR-FCN by a sizeable margin, which validates the superiority of our framework. Besides, we even perform comparably with HOTR and IDN under an inferior experimental setting where HOTR adopts a more advanced transformer encoder-decoder architecture, and both methods are trained with strong supervision. Furthermore, the mAP gap between Rare (training annotations < 10) and Non-rare HOI classes in our results is much smaller than other methods, demonstrating the superior generalization capability of our HOI representation for solving the long-tailed distribution issue. In detail, we achieve a 0.62 mAP gap with ResNet-50

Table 2: Ablation study on HICO-DET dataset. "RN50-FPN(COCO)" denotes the backbone initialized with Faster R-CNN parameters pretrained on MSCOCO dataset while "CLIP RN50" stands for the backbone initialized with CLIP visual encoder. Besides, we construct the knowledge bank $\mathcal{W}_T$ with random initialization, or computing HOI prompts by RoBERTa or CLIP text transformer.

| Methods | Parameter initialization | | CLIP Knowledge | | | SRC | mAP (%) | | |
| | Backbone | knowledge bank | HOI recognition | KTN | score fusion | | Full | Rare | Non-Rare |
|---|---|---|---|---|---|---|---|---|---|
| *baseline* | CLIP RN50 | - | - | - | - | - | 19.52 | 16.58 | 20.40 |
| *Exp 1* | CLIP RN50 | CLIP Text | ✓ | - | - | - | 20.31 | 18.34 | 20.90 |
| *Exp 2* | CLIP RN50 | CLIP Text | ✓(freeze $\mathcal{W}_T$) | - | - | - | 20.09 | 18.23 | 20.64 |
| *Exp 3* | CLIP RN50 | CLIP Text | ✓ | ✓ | - | - | 20.86 | 18.40 | 21.60 |
| *Exp 4* | CLIP RN50 | CLIP Text | ✓ | ✓ | ✓ | - | 22.40 | 20.70 | 22.90 |
| *Exp 5* | CLIP RN50 | - | - | - | - | ✓ | 19.88 | 17.45 | 20.61 |
| *Exp 6* | CLIP RN50 | CLIP Text | ✓ | - | - | ✓ | 20.75 | 19.38 | 21.16 |
| *Exp 7* | CLIP RN50 | CLIP Text | ✓ | ✓ | - | ✓ | 21.53 | 20.05 | 21.97 |
| *ours* | CLIP RN50 | CLIP Text | ✓ | ✓ | ✓ | ✓ | 22.89 | 22.41 | 23.03 |
| *Exp 8* | RN50-FPN (COCO) | - | - | - | - | - | 19.44 | 16.20 | 20.41 |
| *Exp 9* | RN50-FPN (COCO) | random | ✓ | ✓ | ✓ | ✓ | 19.61 | 15.57 | 20.82 |
| *Exp 10* | RN50-FPN (COCO) | RoBERTa | ✓ | ✓ | ✓ | ✓ | 20.45 | 16.46 | 21.65 |

and 1.53 with ResNet-101 backbone, which is much smaller than AlignFormer (3.14) and PPR-FCN (2.64), and supervised methods SSRT (9.01) and GEN-VLKT (4.9).

For **V-COCO** dataset, we report the performance of $AP_{role}$ in both scenario1 and scenario2 for a complete comparison, which are 42.97 / 48.06 $AP_{role}$ with ResNet-50 and 44.74 / 49.97 $AP_{role}$ with ResNet-101 as backbone. As shown in Tab.1, our model achieves significant improvement compared with AlignFormer, and even is comparable with supervised methods TIN and iCAN.

## 4.4 ABLATION STUDY

In this section, we mainly validate the effectiveness of each component with detailed ablation studies on HICO-DET dataset. We use ResNet-50 as the backbone network to reduce experimental costs.

**Baseline:** The baseline adopts the visual encoder from CLIP-RN50 to generate the vanilla HOI representation $v_p$, which is directly used to predict the interaction scores $s_p$. Only pairwise interaction classification loss $\mathcal{L}_p$ is used for model learning.

**HOI recognition:** We augment the *baseline* with a HOI recognition network and observe the full mAP improves from 19.52 to 20.41, as reported in *Exp 1* of Tab. 2. It suggests that the learnable knowledge bank $\mathcal{W}_T$ serves as a powerful classifier to perform image-level HOI recognition and update the visual encoder for better HOI representation. We visualize the learned parameters of knowledge bank in Appendix D to demonstrate its effectiveness. Furthermore, as in *Exp 2*, the performance slightly decreases from 20.31 to 20.09 when we freeze the training of the knowledge bank, indicating that joint learning of visual features and the knowledge bank is more appropriate for HOI detection.

**Knowledge Transfer Network (KTN):** KTN explicitly transfers the CLIP meta-knowledge to pairwise HOI features. As a result, it contributes 0.55 Full mAP improvement (*Exp 3* v.s. *Exp 1*) and most of the performance gains come from Non-rare classes. This result shows KTN is capable of extracting discriminative features from the relational knowledge bank to our HOI representation. We also study the effectiveness of the attention mechanism of KTN in Appendix E.

**Score fusion:** In Tab. 2, we largely improve the Full mAP from 20.86 (*Exp 3*) to 22.40 (*Exp 4*) by fusing the global HOI scores $s_g$ to pairwise interaction score $s_p$. As the HOI recognition network seamlessly inherits the visual-linguistic features from CLIP and directly adopts image labels as supervision, the global interaction scores are pretty accurate and largely enhance the pairwise scores, demonstrating its strong capabilities to cope with long-tailed and fine-grained HOI recognition.

**Self-taught Relatedness Classification (SRC):** Self-taught classification aims to identify the relatedness between human and objects. The improvements from *Exp 4* to *ours* show the effectiveness of our self-taught strategy, which is capable of figuring out the irrelevant human-object pairs and suppressing their interaction scores during inference.

**Combining KTN & SRC:** The ablation results of *Exp 5-7* in Tab. 2 show the KTN and SRC are able to facilitate each other. In detail, the SRC obtains 0.49 Full mAP improvement when the KTN is introduced (*ours* v.s. *Exp 4*), which is only 0.36 without KTN (*Exp 5* v.s. *baseline*). Similarly,

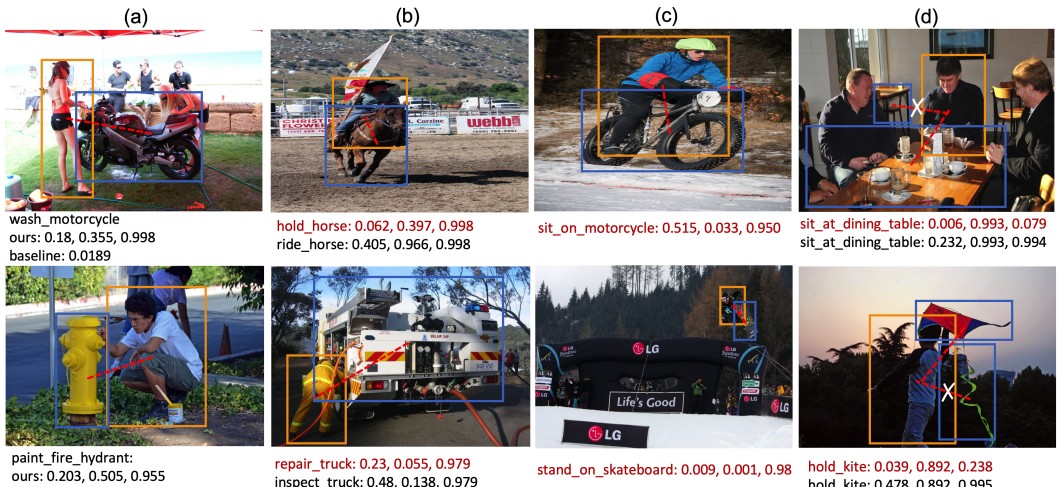

Figure 3: Visualization of HOI detection results on HICO-DET test set. Red scores denote the negative HOI predictions. We mainly demonstrate the model's capabilities on four aspects: **(a)** coping with imbalanced HOI distribution; **(b)** distinguishing subtle differences among interaction types; **(c)** suppressing background HOI classes, and **(d)** pruning irrelevant human-object associations. The numbers reported are normalized pairwise interaction score, global HOI score and relatedness score.

the KTN contributes 0.78 Full mAP improvement with SRC (*Exp 7* v.s. *Exp 6*), which is only 0.55 without SRC (*Exp 3* v.s. *Exp 1*).

**Parameter initialization:** Our visual encoder and knowledge bank are both initialized from CLIP. We also explore different parameter initialization strategy in *Exp 8-10*. Specifically, we initialize the visual encoder with a ResNet50-FPN pretrained on COCO detection task for the baseline (*Exp 8*), and the knowledge bank with random parameters (*Exp 9*) or embeddings of HOI labels from RoBERTa model (*Exp 10*) for the final model. We observe severe drops with all these initialization methods compared with ours, demonstrating the effectiveness and generalization ability of CLIP model. It is worth noting that the mAP of Rare classes decreases from 16.20 in *Exp 8* to 15.57 in *Exp 9*, which suggests the randomly initialized knowledge bank even aggravates the imbalance issue in final model.

### 4.5 QUALITATIVE RESULTS

We show some qualitative results of our method in Fig.3. For each HOI prediction, we report (i) normalized pairwise interaction score, (ii) global HOI score and (iii) relatedness score for *ours*, and only pairwise interaction score for *baseline*. In Fig.3(a), *ours* interaction scores are more confident than *baseline* in Rare HOI classes, demonstrating the generalization ability of our CLIP-guided HOI representation. Besides, when incorporating relational knowledge bank into pairwise HOI representation, our method is capable of distinguishing the subtle differences among similar HOIs in Fig.3(b) (e.g., *repair_truck:0.23* v.s. *inspect_truck:0.48* in the bottom figure). Moreover, in Fig.3(c), the global branch suppresses background HOIs by predicting low global scores for them (e.g., the global HOI score is 0.033 for *sit_on_motorcycle* while the ground-truth is *sit_on_bicycle*). Finally, in Fig.3(d), our self-taught relatedness classification strategy shows strong capability at recognizing the ambiguous human-object associations (e.g., *0.079* v.s. *0.994* in the upper figure).

## 5 CONCLUSION

In this paper, we propose a bi-level knowledge integration strategy that incorporates the prior knowledge from CLIP for weakly-supervised HOI detection. Specifically, we exploit CLIP textual embeddings of HOI labels as a relational knowledge bank, which is adopted to enhance the HOI representation with an image-wise HOI recognition network and a pairwise knowledge transfer network. We further propose the addition of a self-taught binary pairwise relatedness classification loss to overcome ambiguous human-object association. Finally, our approach achieves the new state of the art on both HICO-DET and V-COCO benchmarks under the weakly supervised setting.

## ACKNOWLEDGEMENT

We acknowledge funding from Flemish Government under the Onderzoeksprogramma Artificiele Intelligentie (AI) Vlaanderen programme, Shanghai Science and Technology Program 21010502700 and Shanghai Frontiers Science Center of Human-centered Artificial Intelligence.

## ETHICS STATEMENT

Hereby, we consciously assure that our study is original work which has not been previously published elsewhere, and is not currently being considered for publication elsewhere. We do not have ethics risks as mentioned in the author guidelines.

## REPRODUCIBILITY STATEMENT

We use publicly available benchmarks, HICO-DET and V-COCO, to validate our method. Code is available at https://github.com/bobwan1995/Weakly-HOI.

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

APPENDIX

In this appendix, we first describe the spatial feature generation, and then supplement more experimental results of different CLIP knowledge integration strategies for weakly-supervised HOI detection. For Explanation-HOI (Baldassarre et al., 2020), we further clarify the difference between their mAP evaluation protocol and the standard one. Finally, we demonstrate the limitations, potential negative societal impacts as well as the result error bars of our method.

## A  THE ADVANTAGE OF OUR HOI REPRESENTATION

To verify the improvement obtained with our CLIP-based HOI representation, we visualize the HOI representation $\hat{v}_p$ in feature space with t-SNE(van der Maaten & Hinton, 2008). For clarity, we randomly sample 80 HOI categories, and collect 50 samples for each category. For comparison, we also demonstrate the object-based HOI representation derived from 'Exp 9' in Tab.2 (i.e., the model without CLIP knowledge and using a random knowledge bank). As shown in Fig.4, we observe that CLIP-based HOI representations for different HOI categories are diverse and well separated in feature space, which is better for HOI detection. In contrast, the object-based representations are not well separated in feature space (see the red box region in Fig.4b). Besides, the experimental results in the ablation study (ours v.s. 'Exp 9') also validate the advantage of CLIP-based HOI representation, improving full mAP from 19.61 to 22.89.

## B  ABLATION ON CLIP KNOWLEDGE INTEGRATION

To further demonstrate the superiority of our CLIP knowledge integration strategy, we study several proven techniques for CLIP knowledge transfer in Tab. 3. In *Abl 1*, for each human-object pair, we directly infer the HOI scores with CLIP by computing the cross-modal similarities between their visual union region and the HOI prompts. Without introducing any HOI priors, the promising results indicate the powerful generalization ability of CLIP and motivate the design of incorporating CLIP knowledge for weakly-supervised HOI detection. In *Abl 2*, we duplicate the experiment setting and results from *Exp 8* in Tab. 2 of the main paper. It is a simplified *baseline* model but initializes the visual encoder with a ResNet50-FPN pretrained on COCO detection task. Then we introduce three different CLIP knowledge transfer strategies (*Abl 3-4* and *ours*) based on *Abl 2*.

In *Abl 3*, we directly enhance baseline scores in *Abl 2* with the CLIP similarity scores in *Abl 1* on the inference stage. Without bells and whistles, we obtain 1.12 gain in Full mAP.

Furthermore, in *Abl 4*, we adopt a similar knowledge transfer strategy as GEN-VLKT (Liao et al., 2022), where we initialize the HOI classifier $\mathcal{F}_P$ with HOI prompt and regularize the global HOI representation with CLIP image feature $v_g$. In detail, we first compute the global HOI representation $v_{mean}$ with mean pooling on all pairwise HOI representations, i.e., $v_{mean} = \underset{m}{MeanPool}(\{v_p^m\}_{m=1}^M)$. Here $v_p^m$ is the holistic HOI representation (c.f. Sec. 3.2.3 in the main paper) for $m$-th human-object pair. Then we develop an additional $L2$ loss $\mathcal{L}_{reg}$ to transfer the knowledge from CLIP to HOI representations: $\mathcal{L}_{reg} = L2(v_{mean}, v_g)$. The performance even decreases slightly from 19.44 to 19.39, which might be caused by the incompatibility of parameters between backbone network (ResNet50-FPN pretrained on COCO) and $\mathcal{F}_P$ (HOI prompt embeddings from CLIP). When directly applying the knowledge transfer strategy of GEN-VLKT to a weakly-supervised setting, it is difficult to map the unmatched HOI representation and classification weights to a joint space as the supervisory signals are noisy.

Finally, our approach achieves the best performance compared with other strategies, demonstrating the effectiveness of our bi-level knowledge integration strategy.

## C  SPATIAL FEATURE GENERATION

Following (Zhang et al., 2021c), we generate the spatial feature $v_{sp} \in \mathbb{R}^D$ for each pair of human-object proposals $(\mathbf{x}_h, \mathbf{x}_o)$. Specifically, we first compute the bounding boxes information for $\mathbf{x}_h$ and $\mathbf{x}_o$ separately, including their center coordinates, widths, heights, aspect ratios and areas, all

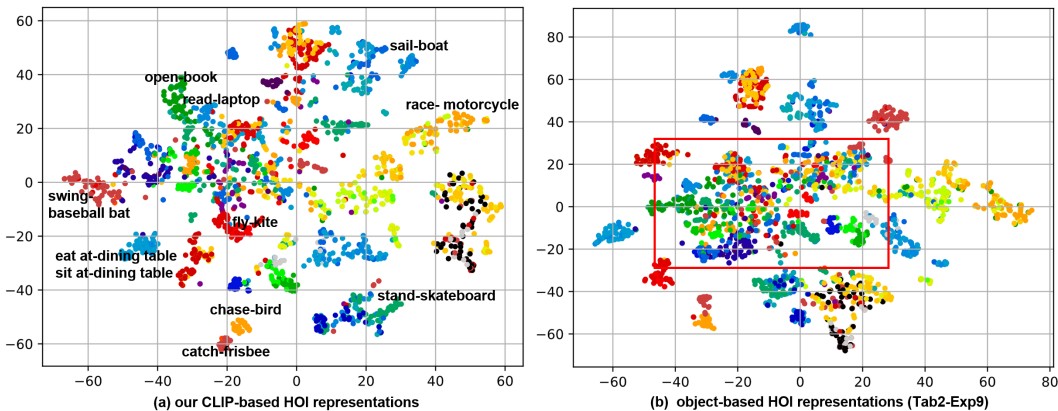

Figure 4: The t-SNE visualization of CLIP-based HOI representation and object-based HOI representation.

Table 3: Ablation of different CLIP knowledge integration strategies on HICO-DET dataset.

| Methods | Experimental setting | mAP (%) | | |
|---------|---------------------|---------|------|----------|
| | | Full | Rare | Non-Rare |
| *Abl 1* | CLIP inference score | 11.84 | 13.72 | 11.27 |
| *Abl 2* | RN50-FPN (COCO) + $\mathcal{F}_P$ random init. | 19.44 | 16.20 | 20.41 |
| *Abl 3* | RN50-FPN (COCO) + $\mathcal{F}_P$ random init. + CLIP inference score | 20.56 | 18.19 | 21.27 |
| *Abl 4* | RN50-FPN (COCO) + $\mathcal{F}_P$ HOI prompt init. + CLIP visual regularization | 19.39 | 15.12 | 20.66 |
| *ours* | CLIP RN50 + HOI recognition + KTN + self-taught relatedness cls. | 22.89 | 22.41 | 23.03 |

normalized by the corresponding dimension of the image. We also encode their relative spatial relations by estimating the intersection over union (IoU), a ratio of the area of $\mathbf{x}_h$ and $\mathbf{x}_o$, a directional encoding and the distance between center coordinates of $\mathbf{x}_h$ and $\mathbf{x}_o$. We concatenate all the above-mentioned preliminary spatial cues and obtain a spatial encoding $\mathbf{p} \in \mathbb{R}^{18}_+$. To encode the second and higher order combinations of different terms, the spatial encoding is concatenated with its logarithm and then embedded to $v_{sp}$: $v_{sp} = \mathcal{F}_{sp}([\mathbf{p}; log(\mathbf{p} + \epsilon)])$. Where $\epsilon > 0$ is a small constant to guarantee the numerical stability, and $\mathcal{F}_{sp}$ is a multi-layer fully connected network.

## D  VISUALIZATION OF HOI KNOWLEDGE BANK $\mathcal{W}_T$

To further understand $\mathcal{W}_T$, we visualize the knowledge bank features initialized by CLIP (Fig.5(a)) and learned from scratch (Fig.5(b)) in feature space by t-SNE. It is worth noting that the knowledge bank learned from scratch is derived from 'Exp 9' in Tab.2. As shown in Fig.5, we observe that the knowledge features of HOI classes initialized with CLIP are more discriminative than random initialized, and show a better clustering result (e.g. the HOI classes in red box regions).

## E  DIFFERENT DESIGNS OF KTN

To further validate the effectiveness of our attention mechanism in KTN, we compare our design with some variants in Tab. 4. First of all, we directly encode the relation-level features within the union region to enhance the pairwise representation rather than the external knowledge bank. As a result, the mAP even decreases a little bit from 20.75 (*Exp 6*) to 20.69 (*Exp 11*). The potential reason is that the union region contains more ambiguous visual relations and background clutters, which are difficult to learn in a weak setting. Besides, we also explore different normalization strategies in KTN. The results in Tab. 4 demonstrate that $Softmax$ operation (*ours*) performs better than uniform attention (*Exp 12*) or $Sigmoid$ operation (*Exp 13*), indicating our attention mechanism is non-trivial and more effective on aggregating the relational cues from HOI knowledge bank.

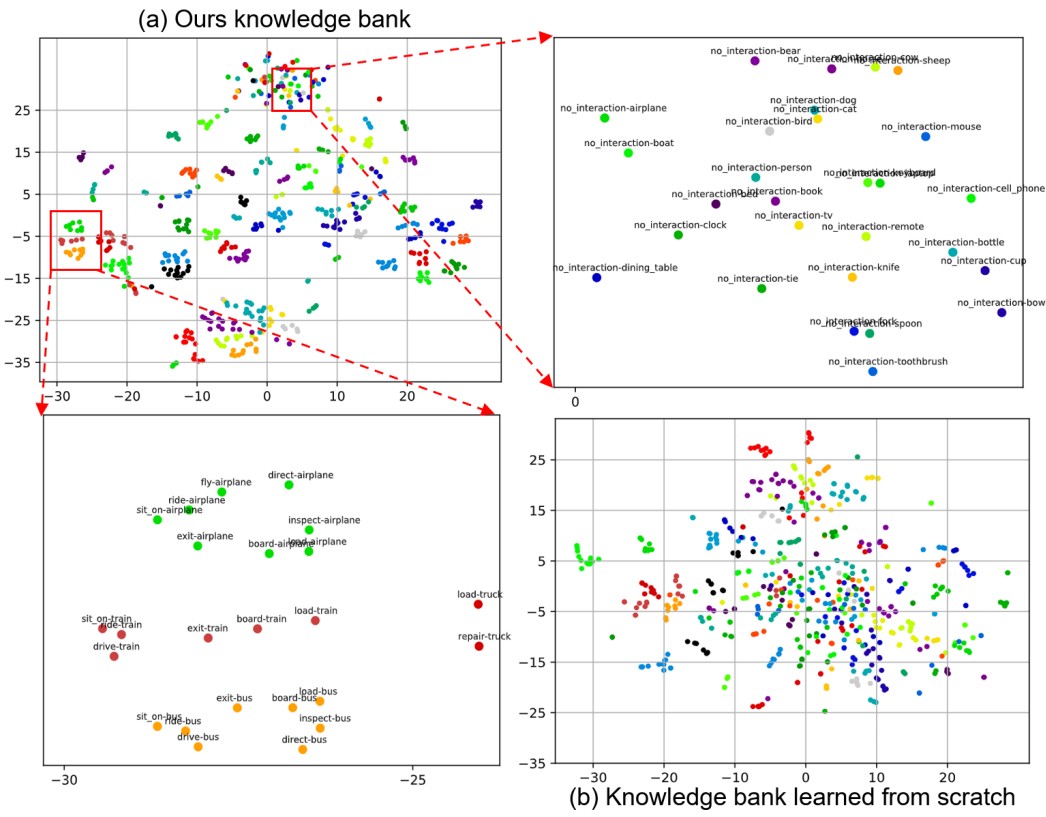

Figure 5: The t-SNE visualization of knowledge bank $\mathcal{W}_T$. (a) is the knowledge bank distribution in feature space based on our CLIP-based HOI representation while (b) is the knowledge bank learned from scratch (the model in Tab.2-Exp 9) based on object-based HOI representation.

Table 4: Different network design of Knowledge Transfer Network (KTN).

| Methods | Parameter initialization | | CLIP Knowledge | | | SRC | mAP (%) | | |
|---|---|---|---|---|---|---|---|---|---|
| | Backbone | knowledge bank | HOI recognition | KTN | score fusion | | Full | Rare | Non-Rare |
| *Exp 11* | CLIP RN50 | CLIP Text | ✓ | ✓(union) | - | ✓ | 20.69 | 19.55 | 21.04 |
| *Exp 12* | CLIP RN50 | CLIP Text | ✓ | ✓(uniform) | - | ✓ | 21.14 | 19.82 | 21.53 |
| *Exp 13* | CLIP RN50 | CLIP Text | ✓ | ✓(sigmoid) | - | ✓ | 21.28 | 19.27 | 21.88 |
| *ours* | CLIP RN50 | CLIP Text | ✓ | ✓ | - | ✓ | 21.53 | 20.05 | 21.97 |

## F TOP-K POSITIVE PAIR SELECTION FOR SRC

In this section we show the results of selecting top-2 and top-5 pairs as positive in Tab. 5. We notice that there is a small performance drop, which is likely to be caused by mislabeling more negative pairs as positive, resulting in model learning with more noise.

## G THE PROMPT GENERATION FOR V-COCO

For the V-COCO dataset, each action has two different semantic roles ('instrument' and 'object') for different objects, like 'cut cake' and 'cut with knife'. We use two different prompt templates to convert a HOI label to a language sentence. For the former one, we take template "a person verb a/an object", and use "a person verb with object" for the latter.

## H EVALUATION METRIC FOR V-COCO

V-COCO dataset has two scenarios for role AP evaluation. In Tab. 1, APS1&2 refer to 'Average Precision in scenario 1&2'. V-COCO dataset has two different annotations for HOIs: the first is a

Table 5: Ablation of top-K positive pair selection for SRC on HICO-DET dataset.

| Methods | mAP (%) | | |
|---------|---------|------|----------|
| | Full | Rare | Non-Rare |
| *Top-5* | 22.45 | 21.61 | 22.70 |
| *Top-2* | 22.49 | 21.83 | 22.69 |
| *ours (Top-1)* | 22.89 | 22.41 | 23.03 |

Figure 6: The screenshot of the evaluation code in Explanation-HOI. (a) is the original code while (b) is the correct one based on the standard evaluation code. We use red rectangle boxes to highlight the most important differences

full label of (human location, interaction type, object location, object type), and the second misses target object (also denoted as 'role' in the original paper (Gupta & Malik, 2015)) annotations, and the label only includes (human location, interaction type). For the second case, there are two different evaluation protocols (scenarios) when taking a prediction as correct [4]: In scenario 1, it requires the interaction is correct & the overlap between the human boxes is $> 0.5$ & the corresponding role is empty, which is more restricted; in scenario 2, it only requires the interaction is correct & the overlap between the person boxes is $> 0.5$.

## I    EVALUATION OF EXPLANATION-HOI

The Explanation-HOI (Baldassarre et al., 2020) has a misunderstanding of mAP evaluation protocol. As shown in Fig.6(a) L200-L205, the Explanation-HOI only takes some specific predicted HOIs into the evaluation process, which has the same HOI labels as groundtruth HOIs. Thus, they ignore lots of false-positive HOI predictions when calculating mAP, leading to an untrustable high mAP score (reported in their original paper). In Fig.6(b) L204-L208, we evaluate all predicted HOIs, which is the same as the standard evaluation protocol proposed in HICO-DET (Chao et al., 2015). The correct results have already been reported in Tab.1 in the main paper.

## J    LIMITATIONS

As described in Sec. 3.1, we adopt an external object detector to generate human-object proposals and then recognize their interactions. Consequently, our method is faced with two limitations brought by erroneous object detection results. Firstly, the positive human-object pairs are not recalled if the human or object proposals are not detected. Secondly, the proposals are kept fixed during learning, which leads to the problem of inaccurate localization and object types.

---

[4]https://github.com/s-gupta/v-coco

# K    RISK OF USING CLIP

For all the methods that adopt CLIP in their model design, there is a potential risk of data leakage as CLIP has seen quite a lot of data during pretraining. For HOI detection task, we cannot get access to CLIP dataset and do not know the exact overlap between CLIP and HOI benchmarks (i.e., HICO-DET and V-COCO), we carefully read Sec. 5 (Data Overlap Analysis) of the CLIP paper (Radford et al., 2021b), including an analysis of the overlap between its dataset with 35 popular datasets (HICO-DET and V-COCO are not included). It shows the overlap is small (median is 2.2% and average is 3.2%) and the influence is limited ("overall accuracy is rarely shifted by more than 0.1% with only 7 datasets above this threshold"). Besides, the training text accompanying an image in the CLIP dataset is often not related to the HOI annotations. Thus, we think the risk is limited.

# L    LICENSE

The licenses of the assets used in our work are listed below, including open-sourced CLIP model, HICO-DET dataset, and V-COCO dataset. As for HICO-DET, we cannot find its license in the paper and the official project page. Thus we provide the official project page instead here for clarity.

1. CLIP: https://github.com/openai/CLIP MIT License
2. VCOCO: https://github.com/s-gupta/v-coco/ MIT License
3. HICO-DET: http://www-personal.umich.edu/ ywchao/hico/ No license

