# OpenReview forum: "Weakly-supervised HOI Detection via Prior-guided Bi-level Representation Learning"
_ICLR.cc/2023/Conference — ICLR 2023 poster_

### Official Review · Reviewer_frLC · 2022-10-23

**Confidence:** 4
**Correctness:** 4
**Technical Novelty And Significance:** 2
**Empirical Novelty And Significance:** 3
**Recommendation:** 6

**Clarity, Quality, Novelty And Reproducibility:**

The proposed approach is detailed in the manuscript and should be able to reproduce without concerns. The author promise to release all codes and trained model for reproducibility purposes.

**Details Of Ethics Concerns:**

N/A.

**Strength And Weaknesses:**

Strength:
+ This work acknowledges the rich information encoded in the pre-trained CLIP model and proposes several modules to better leverage such information for the HOI detection task. This include (1) HOI knowledge bank, and (2) knowledge transfer network.
+ Design a local relation-specific HOI representation that transfer relation-level semantic knowledge from CLIP to pair-wise representations.
+ Design a model-guided relatedness classification loss term (i.e., some form of trustworthiness score) to address the problem in weakly supervised learning.
+ The paper conducted several ablation studies to empirically show the efficacy of the proposed modules. Improvement on rare classes is more significant, demonstrate the benefit of having strong pre-trained model.

Weakness:
- The technical novelty of this work is relatively weak. The relatedness classification loss term is very straightforward and used in earlier work (e.g., using a threshold to remove low-quality prediction). In addition, this work largely benefits from the strong CLIP model (e.g., HOI knowledge bank, visual encoder, and text encoder). Effectively, the long-tailed issue is partially addressed as the encoder is trained with external dataset. The strength is that this work shows how to better leverage the CLIP model in the context of HOI detection. But I would argue that the technical novelty is welcome but limited.

Other concerns:
- Introduction highlights that JOI annotations are prone to image-level labelling errors. Can the author elaborate on how can the model deal with labelling errors if it is presented in training data?
- In Fig 1, please clearly annotate which are the HOI recognition network (is it both local and global branch?)
- In Section 3.2, "The global branch includes a backbone network (Sec.3.2.1) that generates human-
object proposals and extracts image features". The human-object proposals seems to be generated with a pre-trained object detector and provided as an input to global brunch, please clarify this.
- Please comment about the baseline model in ablation study. It seems that the performance is actually higher than the PPR-FCN (which use the same backbone) in Table 1. Is there some clear factors that give the baseline a higher performance?
- Please change "strongly supervised learning" to "supervised learning". The term is well established in the community in the past decades.

**Summary Of The Paper:**

This work addressed the problem of HOI detection tasks under a weakly-supervised learning paradigm. The key inspiration is to leverage the rich vision-language model (i.e., CLIP) to improve the model learning on highly noisy and long-tailed HOI datasets. The key novelties are the novel local relation-specific HOI representation and the relatedness classification terms. The experimental results show that the proposed approach improves the corresponding baseline.

**Summary Of The Review:**

Overall, this paper is well-written and easy to follow. Generally, the work is highly beneficial from the strong visual-language model, where the object semantics are better learned in the pre-trained model. The key contribution of this work is to better exploit CLIP embedding for learning relational information. The designed model is intuitive and provides insights on how to leverage such information for a specific task.

---

### Official Review · Reviewer_q7LZ · 2022-10-28

**Confidence:** 1
**Clarity, Quality, Novelty And Reproducibility:** There is a lack of contribution and n…
**Correctness:** 3
**Technical Novelty And Significance:** 2
**Empirical Novelty And Significance:** 2
**Recommendation:** 5

**Strength And Weaknesses:**

The proposed approach achieves the new state of the art on both HICO-DET and V-COCO benchmarks under the weakly supervised setting.

**Summary Of The Paper:**

This paper proposes a bi-level knowledge integration strategy that incorporates the prior knowledge from CLIP for weakly-supervised HOI detection.

This paper also exploits CLIP textual embeddings of HOI labels as a relational knowledge bank, which is adopted to enhance the HOI representation with an image-wise HOI recognition network and a pairwise knowledge transfer network.

This paper further proposes the addition of a self-taught binary pairwise relatedness classification loss to overcome ambiguous human-object association.



**Summary Of The Review:**

 It lacks any innovative contribution.

---

### Official Review · Reviewer_RZBb · 2022-10-29

**Confidence:** 3
**Clarity, Quality, Novelty And Reproducibility:** Quality, clarity and originality are …
**Correctness:** 3
**Technical Novelty And Significance:** 3
**Empirical Novelty And Significance:** 3
**Recommendation:** 8

**Strength And Weaknesses:**

Strength
- The method achieves the sota performance among weak-supervised hoi detection benchmark.
- Utilizing the prior knowledge from encoded prompt feature bank to enhance the human-object feature is interesting

Weaknesses
- Literature review should be enhanced, e.g., [1].
[1] Weakly Supervised Learning of Interactions between Humans and Objects
- From my humble view,  the score fusion is much better since union region brings lots of noise. Therefore, using the union region feature as  query to get attention score still be noisy.
- It seems that over 50 percent images of HICO—DET only contains one human-object pair (pls correct me if wrong). Therefore, HICO-DET is probably not the best choice to validate idea, especially for ablation study.
- From my prospective, the main contribution of the paper is exploring how to introduce CLIP text prior but it is not strong related to weakly supervised learning. How about the result if taking into account fully supervision?


**Summary Of The Paper:**

For weakly-supervised HOI identification, this paper provides a bi-level knowledge integration technique that integrates the prior information from CLIP. To augment the HOI representation using an image-wise HOI recognition network and a pairwise knowledge transfer network,  the authors specifically use CLIP textual embeddings of HOI labels as a relational knowledge bank. To get over unclear human-object connection, the authors also suggest including a self-taught binary pairwise relatedness classification loss. Finally, in a weakly supervised context, the proposed method surpasses the previous state of the art on the HICO-DET and V-COCO benchmarks.

**Summary Of The Review:**

This paper looks good to me. The main concern is that CLIP text prior seems not to be strongly related to weakly supervised learning considering pseudo labels contribute only a little improvement.

---

### Official Review · Reviewer_qCHU · 2022-10-29

**Confidence:** 2
**Correctness:** 2
**Technical Novelty And Significance:** 2
**Empirical Novelty And Significance:** 3
**Recommendation:** 6

**Clarity, Quality, Novelty And Reproducibility:**

I have problems with the paper presentation (points above in weakness).
The paper is hard to follow. Many notations are used without clear definitions beforehand.

**Details Of Ethics Concerns:**

none as far as i can see

**Strength And Weaknesses:**

[Strength]
The idea of incorporating external knowledge from CLIP is interesting.

[Weakness]
1. The authors promise to release source codes. This would eliminate the issue of reproducibility. However, the paper presentation is bad, which makes readers hard to fully understand and hence, potentially hard to reproduce the model based on the main text.

2. The paper presentation is bad (see more points below).

3. There exists an unfair comparison with baselines in the experiments (see elaborations below).

I elaborated on the specific points below:
[clarity]
The paper presentation is very bad. I have trouble understanding some parts of the model designs and the problem setting. The authors are strongly encouraged to explicitly list out what are the supervised signals in the problem setting in both fully supervised and weakly supervised cases. Please clearly define every math notation before using them. I list out several key confusions:

1.1. What is the detection score s_h defined in sec 3.1? Ground truth bounding box coordinates? How many scores are there?

1.2. Define relatedness and interaction scores in the introduction.

1.3. Add vg defined in sec3.2.1 to fig1. Also including several other key variables in Fig1 would help readers understand the designs better.

1.4. I have trouble understanding this relatedness score. Up to 3.2.5, there is no definition of sb. What is the relatedness score exactly? Is this binary score given as ground truth? If so, the authors should spell it out at the beginning of the paper.

1.5. In sec3.3, it talks about self-taught relatedness classification loss. What is exactly given in this case? If binary cross-entropy loss is used and bm is given, this relatedness acts as the ground truth and it should be explicitly introduced in the problem setting in Sec 3.1.

1.6. what is given in the weakly supervised setting? What are the image-level annotations only? Go back to the tuples O defined in Sec3.1, what is exactly missing or not given in a weakly supervised setting?


1.7. In table1, what are APS1 and S2 in the rightmost column?

[other questions]
2. I am familiar with scene graph generation. HOI seems to be a subset of that problem. Would this weakly supervised method generalize to scene graph generation?

3. I found the comparisons of methods are unfair and the results are puzzling. In table 1, baselines are using different backbones and object detectors. If so, it is very hard to say whether the performance improvement is simply because of the change of backbones and detectors, or because the proposed weakly supervised methods are efficient.

4. Utilizing the external knowledge from CLIP is an interesting idea! However, would this put other methods at disadvantage as CLIP has been presented with more data than existing methods?

**Summary Of The Paper:**

The paper presents a new weakly supervised method in the human-object interaction task. The method exploits the prior knowledge from CLIP.

**Summary Of The Review:**

The paper presentation is bad which creates troubles for me to fully understand the method section.
This subsequently creates problems for me to appreciate the novelty of the method.
I am also having issues with the baseline comparisons. I do not think it is fair to some extent (see elaborations above).

I would vote for REJECT. However, it is possible that I might revise my ratings based on the authors' feedback after the rebuttal period.

---

### Official Review · Reviewer_S5GP · 2022-10-30

**Confidence:** 5
**Correctness:** 4
**Technical Novelty And Significance:** 3
**Empirical Novelty And Significance:** Not applicable
**Recommendation:** 8

**Clarity, Quality, Novelty And Reproducibility:**

The approach is new and interesting. It could be useful for weakly-supervised learning.
The implementation details are useful for reproduction.

**Strength And Weaknesses:**

Strength:
1.	The motivation is clear. Applying CLIP in weakly supervised learning and facilitating the learning of an HOI detector is essential.
2.	The approach is interesting. Though combining the detector with CLIP has been studied, but combining them for weakly-supervised learning has been less explored.
3.	The experimental study is promising and comprehensive. The visualizations are also provided.
4.	The ablation study is in detail.

Weakness:
1.	Regarding the whole framework, which part is vital for using CLIP to guide weakly supervised learning? I think the discussion is necessary (but I didn’t find clear answer in the discussion) and help this paper to be distinguished from the other related work.
2.	The knowledge bank is based on classes appearing in the full dataset and is defined by the text. Can you explain how the size of the knowledge bank affects performance? After all, when there exists a brunch of interaction classes, I am not sure about the training efficiency and workload.
3.	SRC only does not help much with the detection, according to Table 2. It is not very effective and is kind of counterintuitive. I would recommend providing a more detailed explanation or removing the SRC part.
4. When a CLIP model is used, it is always necessary to explain the potential issue of fair comparison. After all, CLIP has seen quite a lot of training data during pretraining, and there is a risk of potential data leakage. As such, explanation is necessary.


**Summary Of The Paper:**

This paper targets on HOI detection in a weakly supervised manner (i.e., only image description is provided, no instance-level annotation). To improve the generalization towards fine-grained human-object interactions and mitigate the biased learning caused by imbalanced data distribution, this paper proposes a training strategy to distill the knowledge from CLIP model. Specifically, they incorporate CLIP in HOI at both the image and the instance levels. At the image level, they generate a global HOI knowledge bank and predict image-level HOI descriptions. At the instance/object level, the attention mechanism is employed to enrich the representation of HOI features and then improve the performance. A comprehensive evaluation is conducted and a detailed analysis is provided.

**Summary Of The Review:**

The overall presentation of this paper is good. However, a few more discussions should be provided. Please see details in weakness.

---

### Decision · Program_Chairs · 2023-01-20

**Decision:**

Accept: poster

**Justification For Why Not Higher Score:**

The reviewers recommending acceptance comment that they prefer to rate the paper a 7 rather than 8, so the paper is clearly above the bar, but enthusiasm for the ideas presented is not sufficient for oral presentation recommendation.

**Justification For Why Not Lower Score:**

No reviewer recommends rejection.

**Metareview: Summary, Strengths And Weaknesses:**

The paper proposes to utilize prior knowledge from CLIP, e.g. by pruning human-object associations from pretrained models. Reviewers appreciate the idea (especially in the context of weakly-supervised learning), the results and ablations, and the writing. They raise concerns about parts of the method, aspects of evaluation (e.g. data leakage), and novelty. For most reviewers, the concerns are well addressed by the rebuttal.

**Note From Pc:**

if the above contains the word "oral" or "spotlight" please see: "oral" presentation means -> notable-top-5% and "spotlight" means -> notable-top-25%. As stated in our emails, we are disassociating presentation type from AC recommendations